# Quantitative Trait Loci (QTL) Analysis of Seed Protein and Oil Content in Wild Soybean (*Glycine soja*)

**DOI:** 10.3390/ijms24044077

**Published:** 2023-02-17

**Authors:** Woon Ji Kim, Byeong Hee Kang, Chang Yeok Moon, Sehee Kang, Seoyoung Shin, Sreeparna Chowdhury, Man-Soo Choi, Soo-Kwon Park, Jung-Kyung Moon, Bo-Keun Ha

**Affiliations:** 1Department of Applied Plant Science, Chonnam National University, Gwangju 61186, Republic of Korea; 2BK21 FOUR Center for IT-Bio Convergence System Agriculture, Chonnam National University, Gwangju 61186, Republic of Korea; 3National Institute of Crop Science, Rural Development Administration (RDA), Wanju 55365, Republic of Korea

**Keywords:** wild soybean, protein, oil, quantitative trait loci, indels

## Abstract

Soybean seeds consist of approximately 40% protein and 20% oil, making them one of the world’s most important cultivated legumes. However, the levels of these compounds are negatively correlated with each other and regulated by quantitative trait loci (QTL) that are controlled by several genes. In this study, a total of 190 F_2_ and 90 BC_1_F_2_ plants derived from a cross of Daepung (*Glycine max*) with GWS-1887 (*G. soja*, a source of high protein), were used for the QTL analysis of protein and oil content. In the F_2:3_ populations, the average protein and oil content was 45.52% and 11.59%, respectively. A QTL associated with protein levels was detected at Gm20_29512680 on chr. 20 with a likelihood of odds (LOD) of 9.57 and an R^2^ of 17.2%. A QTL associated with oil levels was also detected at Gm15_3621773 on chr. 15 (LOD: 5.80; R^2^: 12.2%). In the BC_1_F_2:3_ populations, the average protein and oil content was 44.25% and 12.14%, respectively. A QTL associated with both protein and oil content was detected at Gm20_27578013 on chr. 20 (LOD: 3.77 and 3.06; R^2^ 15.8% and 10.7%, respectively). The crossover to the protein content of BC_1_F_3:4_ population was identified by SNP marker Gm20_32603292. Based on these results, two genes, *Glyma.20g088000* (S-adenosyl-l-methionine-dependent methyltransferases) and *Glyma.20g088400* (oxidoreductase, 2-oxoglutarate-Fe(II) oxygenase family protein), in which the amino acid sequence had changed and a stop codon was generated due to an InDel in the exon region, were identified.

## 1. Introduction

Soybean (*Glycine max* L.) is an important crop worldwide that represents a major source of protein and vegetable oil for the human diet and animal feed. As of 2021, soybean was the largest source of protein meal in the world, at 243.6 t, and the second-largest source of vegetable oil (58.7 t) after palm oil (http://soystats.com/, accessed on 12 February 2023). In Asian countries, soybean seeds are used to produce a number of food products, including soymilk, tofu, soybean paste, natto, and soy sauce. In the West, soybean is typically used for soybean meal and seed oil. Soybean seeds generally consist of 40% protein and 20% oil [1], and these traits are negatively correlated with each other [2,3]. For this reason, it is very difficult to improve both traits simultaneously. In addition, because there is a negative correlation between seed yield and protein content [4], high-protein varieties need to be developed with care. In addition to being easily influenced by environmental factors, the protein and oil content of soybean seeds is regulated by the polygenes and quantitatively inherited [5,6]. These polygenes can be divided into major genes, which are less influenced by the environment and that have a significant influence on these levels; and minor genes, which have a weaker influence.

Wild soybean (*G. soja* Sieb. and Zucc.), the ancestor of cultivated soybean, has high genetic diversity and is thus valuable as a breeding material for soybean breeding programs [7,8]. Various studies have used wild soybean to improve biological stress resistance, abiotic stress tolerance, nutrition, and yields [9]. The average protein content of wild soybean is reported to be higher than that of cultivated soybean, although this may be due to correlations with the yield or oil content [9]. In the study by Chen and Nelson (2004), the protein content of wild and cultivated soybean lines was about 47% and 40%, respectively, while the oil content was 15% and 11%, respectively [10].

After the publication of the soybean genome for the first time by Schmutz et al. (2010) [11], Ha et al. (2012) [12] advanced genomic research further with the integration of physical maps for *G. max* and *G. soja*. QTL mapping uses F_2_, backcross (BC), and recombinant inbred line (RIL) populations derived from bi-parental crosses. In many soybean populations, the QTLs for proteins and oils have been mapped to genomic regions on chromosomes 15 and 20 [13,14,15,16,17]. Major QTLs for protein and oil content were identified by Diers et al. (1992) using RFLP markers for the F2 population through the crossbreeding of *G. max* (A81-356022) and *G. soja* (PI 468916) [18]. The QTL located on chromosome 15 has been fine-mapped at an interval of 535 kb between simple sequence repeat (SSR) markers (Kim et al. 2016) [19], while the candidate gene for the QTL located on chromosome 20 has been cloned as *Glyma.20G85100* encoding the CCT domain [20].

To date, 255 and 322 protein and oil content-related QTLs, respectively, have been identified using bi-parental populations (https://www.soybase.org/, accessed on 12 February 2023). However, these QTLs may include multiple duplicate detections, so the Soybean Genetics Committee has emphasized the importance of experimentally confirming QTLs and adding “*cq*” in front of the original QTL name to indicate that it has been confirmed [20,21]. In total, 16 QTLs each have been confirmed for protein and oil content (https://www.soybase.org/), and these are distributed across 11 chromosomes, including chromosomes 15 and 20 [18,19,20]. Of these, the only QTLs derived from wild soybean are *cqpro*-003 and *cqoil*-004 [22]. Therefore, the purpose of this study was to discover new genes for protein and oil content from wild soybean using two progeny types derived from high-protein wild soybean lines.

## 2. Results

### 2.1. Phenotypic Variation in the Protein and Oil Content

In the present study, the seed oil and protein content were measured using seeds harvested from F_3_ and BC_1_F_3_ progeny lines in 2019 and 2020, respectively. The protein content of Daepung and GWS-1887, the parents of the F_2:3_ line, in 2019 was 37.10% and 50.37%, respectively, while the oil content was 19.81% and 5.83%, respectively. In the 190 F_2:3_ plants, the protein content ranged from 40.08% to 50.96% with a mean of 45.52%, and the oil content ranged from 7.84% to 16.61% with a mean of 11.59% (Table 1).

The protein content of Daepung and GWS-1887, the parents of the BC_1_F_2:3_ line, in 2020 was 40.05% and 49.28%, respectively, and the oil content was 16.53% and 5.34%, respectively. In the BC_1_F_2_ population, the protein content ranged from 31.50% to 49.54% with a mean of 44.25% and the oil content ranged from 7.73% to 14.84% with a mean of 12.14% (Table 2).

The phenotypic variation of the F_2_ and BC_1_F_2_ populations followed a normal distribution (Figure 1 and Figure 2, respectively).

### 2.2. Linkage Maps and QTL Analysis

Linkage maps for the F_2_ and BC_1_F_2_ populations were constructed using polymorphic SNP markers acquired from SoySNP6K Illumina BeadChips (Appendix A, respectively). In the F_2_ population, 2592 polymorphism markers were used, with an average chromosome length of 95 cM and an average of 130 markers located across each of the 20 chromosomes (Table 3).

In the BC_1_F_2_ population, 1063 polymorphism markers were used, with an average chromosome length of 60 cM (except chromosome 12, which did not show polymorphism) and an average of 130 markers located on each of the 19 chromosomes (Table 4).

The average genetic interval for both the F_2_ and BC_1_F_2_ populations was 1.1 cM. QTLs for the protein and oil content in both populations were identified using MQM mapping analysis. In the F_2:3_ population, QTLs for protein and oil content were identified on chromosomes 20 and 15, respectively (Figure 3), and these QTLs accounted for 17.2% and 12.2% of the phenotypic variation, respectively, with additive effects of the alleles on these traits of −1.10 and 0.59 (Table 5).

On the other hand, in the BC_1_F_2:3_ population, QTLs for protein and oil content were both identified on chromosome 20 (Figure 4), accounting for 15.8% and 10.7% of the phenotypic variation, respectively, with additive effects of the alleles on these traits of −1.49 and 0.62 (Table 6).

### 2.3. Crossover Detection

From the BC_1_F_3_ population, two lines with a heterozygote at a position expected to represent a high-protein-promoting gene on chromosome 20 were selected and advanced to produce a BC_1_F_3:4_ generation, followed by genotyping and protein content measurement. In Table 7, crossover occurred at 33,049,242 bp, and the individual with the genotype of the parent Daepung had average protein levels of 44.10 g, while individuals with the genotype of the parent GWS-1887 had a protein level of 47.58 g. In addition, crossover occurred at 32,603,292 bp, and individuals with the genotype of the parent Daepung had average protein levels of 45.25 g, and the individual with the genotype of the parent GWS-1887 had a protein level of 48.54 g. Given the genotypes of the two lines, it was predicted that the range of genes related to high protein levels is present at least downstream of position 32,603,292 bp.

### 2.4. Genome Re-Sequencing

Genome-resequencing analysis was conducted on wild soybean GWS-1887, which has a high protein content. The total number of sequencing reads was about 260 million with a sequencing depth of 38.6× and a total size of about 39 billion bp, while the coverage for the reference genome was 95.3%. Compared with the Williams 82 reference genome, approximately 4.7 million SNPs and 0.9 million InDels were identified in GWS-1887 (Table 8).

A total of 21 protein-related candidate genes with InDels were detected between Gm20_27578013, which is a molecular marker identified as a result of QTL mapping in the BC_1_F_2:3_ population, and Gm20_32603292, which was identified as the crossover site in BC_1_F_3:4_ (Table 9).

Of these, large InDels were identified in five genes (*Glyma.20G085300*, *Glyma.20G085450*, *Glyma.20G085800*, *Glyma.20g087000*, and *Glyma.20G088000*), small InDels were most common in 40 loci in *Glyma.20G085300*, and three large InDels were present in *Glyma.20G088000*. In particular, InDels occurred in the exon region of genes *Glyma.20g088000* and *Glyma.20g088400*, and the InDels of the two genes generated stop codons with amino acid frameshifts (Figure 5).

Of these, the stop codon in *Glyma.20g088000* is expected to greatly simplify the structure of the protein (Figure 6).

## 3. Discussion

QTL analysis of the protein and oil content in soybean has been well-studied in previous research. The present study searched for QTLs related to high protein content using two F_2_ and BC_1_F_2_ populations derived from a cross between cultivated soybean variety Daepung and wild soybean variety GWS-1887. The protein content of cultivated soybean is known to be about 40% [1], whereas wild soybean GWS-1887 has protein levels close to 50% [9,10,23]. This suggests that wild soybean GWAS-1887 may be useful for QTL analysis in terms of mapping the genetic regions associated with high protein content in soybean. However, the crossbreeding between *G. max* and *G. soja* may lead to linkage drag and consequent negative introgression such as reduced yields [9]. Daepung exhibited a higher annual variation in its protein content (37.10% in 2019 and 40.05% in 2020) than did GWS-1887 (50.37% in 2019 and 49.28% in 2020) (Table 1 and Table 2). Several studies have reported that a lack of soil moisture reduces the protein content of soybean [24,25,26]. The rainfall in the experimental area in August 2019 during the soybean development period was lower than normal, while the rainfall in August 2020 was above average (http://www.kma.go.kr/, accessed on 12 February 2023). Therefore, it appears that the protein content of wild soybean has a higher environmental stability than cultivated soybean.

Previous QTL analysis of the protein and oil content in soybean seeds has been conducted with various populations, with the identified QTLs distributed across 20 chromosomes (https://www.soybase.org/). Of these, major QTLs for protein and oil content are present on chromosomes 15 and 20 [18], and several researchers have attempted to narrow down their precise location [13,19,20,21,22,27]. In the present study, the QTLs related to protein and oil content in the F_2:3_ population were identified as Gm20_29512680 and Gm15_3621773, respectively, whereas in the BC_1_F_2:3_ population, marker Gm20_27578013 was identified for both protein and oil. Kim et al. (2016) reported the fine-mapping of a backcross line of Williams 82 and PI 407788A with 96 BARCSOYSSR markers and found that the QTL related to the protein and oil content on chromosome 15 was present in a 535 kb region from the physical position 3.59 Mbp to 4.12 Mbp [19]. These results are consistent with the SNP marker Gm15_2621773 at the physical position 3.63 Mbp detected for oil content in the F_2:3_ population in the present study. Recently, cqSeed protein-003 located on chromosome 20 was narrowed down through fine-mapping to a 77.8 kb region between genetic marker BARCSOYSSR_20_0670 and BARCSOYSSR_20_0674 (31.74 to 31.82 Mbp), and the *Glyma.20G85100* gene encoding the CCT domain was identified as a candidate gene involved in protein content [20].

In our results, protein-related QTLs were mapped to Gm20_29512680 at 30.61 Mbp and Gm20_27578013 at 28.69 Mbp on chromosome 20 in the F_2:3_ and BC_1_F_2:3_ populations, respectively. These results are consistent with the 24.55–32.91 Mbp range reported by Bolon et al. (2010) [15] and the 28.7–31.1 Mbp range reported by Hwang et al. (2014) [13], but not with the 32.71–33.08 Mbp range identified by Vaughn et al. (2014) [14] and the 31.74–31.82 Mbp range found by Fliege et al. (2022) [20]. The reason for these inconsistencies could the low LD with the surrounding markers [20], and it is known that the wild soybean variety used as a parent in this study has a lower LD than cultivated soybean [28]. For more accurate confirmation of the location, crossovers around the QTL detected in BC_1_F_2:3_ were identified but could not be found, and it was concluded that a crossover occurred at markers Gm20_33049242 and Gm20_32603292 in the two BC_1_F_3:4_ lines, which was advanced one generation by selecting high-protein lines. Therefore, it was predicted that the protein-related gene is present in the region downstream of Gm20_32603292.

Based on the results of QTL mapping, InDels were then searched for in the candidate genes located at around 30 Mbp between the Williams 82 reference genome and GWS-1887. Interestingly, no mutation was detected in the Glyma.20G85100 gene of the CCT motif family protein, which was recently cloned as a major protein-related gene [20]. These results suggest that other major protein-related genes may be present in a similar genetic region. Wang et al. (2021) selected protein-related candidate gene *Glyma.20g088000* using DEG analysis via RNA-seq, and it was found that *Glyma.20g088000* had a significant difference in its sequence between the high-protein Nanxiadou 25 and low-protein Tongdou 11 varieties due to InDels [16]. Interestingly, *Glyma.20g088000* (S-adenosyl-l-methionine-dependent methyltransferase) was also selected as a candidate gene in the present study because small and large InDels occurred within several regions of the gene. In addition, *Glyma.20g086900* (aldehyde dehydrogenase-related) and *Glyma.20g088400* (oxidoreductase, 2-oxoglutarate-Fe(II) oxygenase family protein) genes were selected by Lee et al. (2019) as a result of a GWAS for the soybean seed protein content from maturation groups I to IV [17]. These two genes were also identified as candidate genes in this study.

In the present study, it was confirmed that *Glyma.20g088000* and *Glyma.20g088400* had a large InDel in the 5′ first exon and a small InDel in the 3′ third exon, respectively (Figure 5). Nonsense mutations that create stop codons and frameshifts in which amino acids are rearranged can disrupt the function of a gene [29]. In one example, truncated polypeptides generated as a result of nonsense mutations resulted in the loss of anthocyanin pigments associated with the color of soybean flowers [30]. In particular, the stop codon in *Glyma.20g088000* is expected to greatly simplify the structure of the protein (Figure 6), thus it is likely to have a significant effect on the expression of its function. Although these candidate genes have potential functions in metabolism, the mechanisms of how they relate to seed composition require further study. In addition, the results collectively suggest that protein content may be regulated by the complex interaction of multiple genes located at around 30 Mbp on chromosome 20.

## 4. Materials and Methods

### 4.1. Plant Materials

In the present study, 180 F_3_ and 90 BC_1_F_4_ populations derived from a cross between Daepung and GWS-1887 were analyzed. Daepung, which was used as the female, recurrent parent, is an elite Korean variety that is strongly resistant to disease and shattering and has high yields [31], while GWS-1887, which was used as the male parent, was selected from the core collection of wild soybean accessions from the Rural Development Administration (RDA) because it has a protein content of 50% or higher. In the summer of 2018, F_1_ seeds were obtained from artificial crossbreeding in an experimental field at Chonnam University (Gwangju, 36°17′ N, 126°39′ E, Republic of Korea). The F_1_ seeds were planted in a greenhouse during the 2018–2019 winter season to obtain F_2_ seeds, with the generation then advanced from F_2_ to F_3_ in the summer of 2019. At the same time, F_1_ seeds were backcrossed in the summer of 2019 to obtain BC_1_F_1_ seeds. The produced BC_1_F_1_ seeds were planted in a greenhouse during the 2019–2020 winter period to obtain BC_1_F_2_ seeds. Finally, in the summer of 2020, the BC_1_F_2_ generation was advanced and BC_1_F_3_ seeds were obtained.

### 4.2. Analysis of Protein and Oil Content

All harvested seed samples were dried at 40 °C for 7 d and then pulverized using a coffee grinder to produce 3 g each for subsequent analysis. The crude protein content was measured using the Kjeldahl method. Reagents required for digestion, distillation, and titration were prepared, including 0.1N hydrochloric acid, a decomposition accelerator (containing 10 g of potassium sulfate and 1 g of copper sulfate), 40% sodium hydroxide solution, and 1% boric acid solution with 100 mL and 70 mL of Bromocresol green and methyl red solutions, respectively. The sample solution was prepared by mixing 0.7–1.0 g of the ground seed and 7–8 g of the decomposition accelerator with 10 mL of sulfuric acid in a decomposition bottle. Digestion was carried out by heating the sample solution at a slow ramping rate until no visible bubbles remained and the solution became transparent. The solution was then analyzed using a Kjeltec 1030 Autoanalyzer (FOSS Tecator AB, Hogans, Sweden) following the manufacturer’s instructions.

The crude oil content was measured using ether extraction. For this, an oil metering bottle was pre-dried at around 95–100 °C for 2 h followed by cooling in a desiccator for 30 min. Following this, 2–3 g of the sample wrapped in No. 2 filter paper was dried at the same temperature and for the same duration of time as the oil metering bottle. After drying, the sample was placed in a Soxtec 1043 instrument (FOSS Tecator AB, Hogans, Sweden), and subjected to a flow of ether at 80 °C for 8 h to extract the oil. The processed ether was then collected in an oil metering bottle and subsequently dried (95–100 °C for 3 h) followed by cooling in a desiccator (40 min) and weighing. The oil content was determined by subtracting the weight of the empty oil metering bottle from the weight of the bottle containing the extract.
$$Crude\;oil\left(\%\right)=\frac{\mathrm{Oil}\;\mathrm{bottle}\;\mathrm{weight}\;\mathrm{after}\;\mathrm{extraction}-\mathrm{Oil}\;\mathrm{bottle}\;\mathrm{weight}\;\mathrm{before}\;\mathrm{extraction}}{\mathrm{Sample}\;\mathrm{weight}}\times 100$$

### 4.3. DNA Extraction and SNP Genotyping

Fresh leaf tissue was collected at the beginning of growth for DNA extraction and ground using liquid nitrogen in a mortar. Genomic DNA was isolated from 20 mg of lyophilized leaf tissue using a DNeasy Plant Mini Kit (QIAGEN, Valencia, CA, USA) according to the manufacturer’s protocol. The quality and quantity of the extracted total DNA were verified using a Nano-MD UV-Vis spectrophotometer (Scinco, Seoul, Republic of Korea). The extracted DNA was stored in a freezer at −80 °C until further use. A total of 270 samples, consisting of 180 F_2_ and 90 BC_1_F_2_ plants and two replications of each parental plant (Daepung and GWS-1887) were genotyped using a SoySNP6K Illumina BeadChip (Illumina, San Diego, CA, USA) at TNT Research Co. (Anyang, Republic of Korea). The SNP alleles were called using Illumina’s GenomeStudio (Illumina, Inc., San Diego, CA, USA).

### 4.4. Genetic Linkage Analysis

A genetic linkage map was constructed using the Kosambi mapping function in Joinmap v4.1 (Kyazma, Wageningen, The Netherlands). For genetic analysis, MQM mapping was employed with MapQTL 6.0 (Kyazma, Wageningen, The Netherlands). In the F2 population, permutations were conducted to determine the genome-wide significance threshold for the LOD scores, with the number of permutations set at 1000. In the BC1F2 population, an LOD score of ≥3.0 was set as the threshold for determining the presence of a QTL. LOD graphs and the location maps for the QTLs were created with MapChart2.2.

### 4.5. Re-Sequencing

Re-sequencing analysis was commissioned by Insilicogen (Yongin, Republic of Korea) and performed using an Illumine Novaseq 6000 platform. A library was constructed from DNA fragments with 151 bp paired ends read using a DNA Sample Prep Kit (Illumina) following the manufacturer’s instructions. An analysis pipeline for detecting mutations in the sequencing data for the entire genome was employed with an NF-Core/SAREK workflow [32]. The snpEff tool was used for genetic variation annotation and effect prediction, while the snpEff database was referenced to Glycine max var. Williams 82 [11]. The whole genome sequencing data of GWS-1887 were deposited in NCBI under the BioProject PRJNA915129.

## 5. Conclusions

In this study, QTL mapping of the protein and oil content in soybean seeds was conducted using two progeny populations derived from high-protein wild soybean lines. QTL was detected in the region of *cqPRO*-003, which has been previously reported as a major QTL related to protein content, but as a result of resequencing, no difference was observed from the recently cloned candidate gene *cqPRO*-003. On the other hand, new candidate genes *Glyma.20g088000* and *Glyma.20g088400*, which contained InDels, were discovered. This suggests that the protein content may be regulated by the complex interaction of multiple genes and associations other than those that have previously been reported.

## Figures and Tables

**Figure 1 ijms-24-04077-f001:**
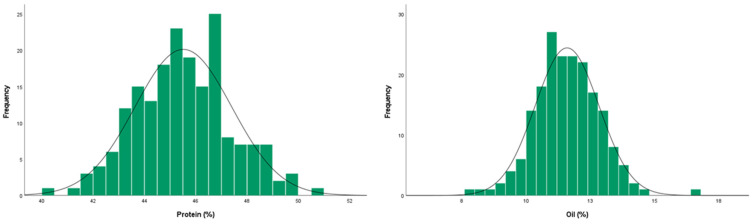
Distribution of the seed protein and oil content in the F_2:3_ mapping population derived from a cross between Daepung and GWS-1887 in 2019.

**Figure 2 ijms-24-04077-f002:**
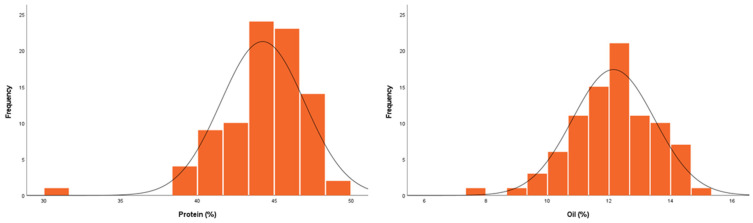
Distribution of the seed protein and oil content in the BC_1_F_2:3_ mapping population derived from a cross between Daepung and GWS-1887 in 2020.

**Figure 3 ijms-24-04077-f003:**
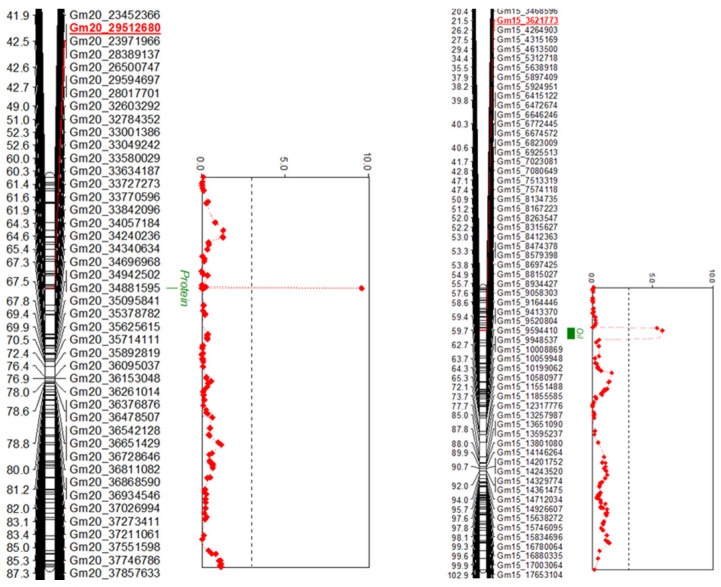
Likelihood of odds (LOD) plot for the seed protein and oil content using an LOD threshold of 3.0 (the vertical dotted line). These QTLs were mapped in the F_2:3_ population derived from a cross between Daepung and GWS-1887.

**Figure 4 ijms-24-04077-f004:**
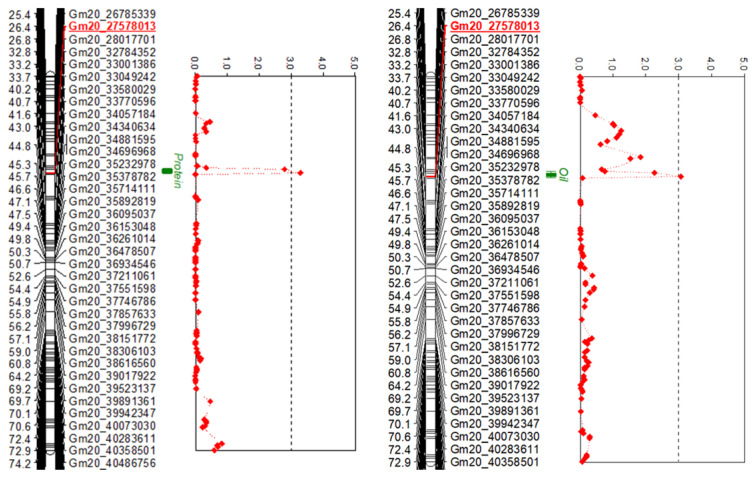
Likelihood of odds (LOD) plot for the seed protein and oil content using an LOD threshold of 3.0 (the vertical dotted line). These QTLs were mapped in the BC_1_F_2:3_ population derived from a cross between Daepung and GWS-1887.

**Figure 5 ijms-24-04077-f005:**
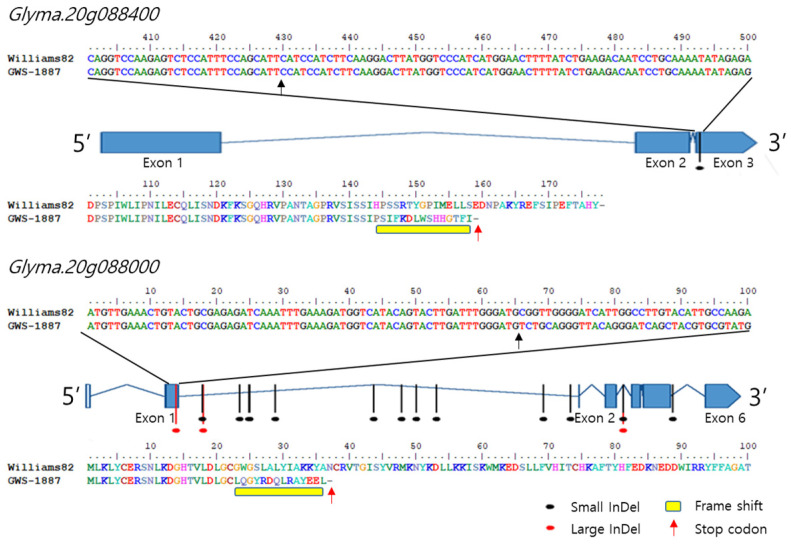
InDels in the *Glyma.20g088400* and *Glyma.20g088000* genes from GWS-1887.

**Figure 6 ijms-24-04077-f006:**
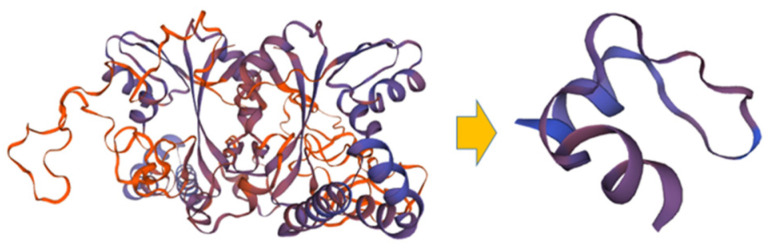
3D prediction of the structure of the *Glyma.20g088000* protein from GWS-1887.

**Table 1 ijms-24-04077-t001:** Seed protein and oil content of the F_2:3_ mapping population and its parental lines Daepung and GWS-1887 in 2019.

	Daepung	GWS-1887	Daepung × GWS-1887 F_2:3_				
			Min	Max	Average	Skew	Kurt
Protein (%)	37.10	50.37	40.08	50.96	45.52	0.08	0.04
Oil (%)	19.81	5.83	7.84	16.61	11.59	0.15	1.04

Min, minimum; Max, maximum; Skew, skewness; Kur, Kurtosis.

**Table 2 ijms-24-04077-t002:** Seed protein and oil content of the BC_1_F_2:3_ mapping population and its parental lines Daepung and GWS-1887 in 2020.

	Daepung	GWS-1887	Daepung × GWS-1887 BC_1_F_2:3_				
			Min	Max	Average	Skew	Kurt
Protein (%)	40.05	49.28	31.50	49.54	44.25	−1.32	4.40
Oil (%)	16.53	5.34	7.73	14.84	12.14	−0.47	0.55

Min, minimum; Max, maximum; Skew, skewness; Kur, Kurtosis.

**Table 3 ijms-24-04077-t003:** Summary of the genetic linkage map for the F_2_ mapping population derived from a cross between Daepung and GWS-1887.

Chr.	Marker No.	Length (cM)	Average Spacing (cM)	Chr.	Marker No.	Length (cM)	Average Spacing (cM)
1	105	129.0	1.2	11	142	154.2	1.1
2	110	168.4	1.5	12	106	134.3	1.3
3	130	146.7	1.1	13	165	161.1	1.0
4	92	131.1	1.4	14	138	126.6	0.9
5	92	126.8	1.4	15	134	141.5	1.1
6	140	173.7	1.2	16	110	113.7	1.0
7	158	141.5	0.9	17	131	138.6	1.1
8	182	181.6	1.0	18	140	136.9	1.0
9	113	133.6	1.2	19	138	151.0	1.1
10	153	158.2	1.0	20	116	148.9	1.3
				Total	2595	2897.4	1.1

**Table 4 ijms-24-04077-t004:** Summary of the genetic linkage map for the BC_1_F_2_ mapping population derived from a cross between Daepung and GWS-1887.

Chr.	Marker No.	Length (cM)	Average Spacing (cM)	Chr.	Marker No.	Length (cM)	Average Spacing (cM)
1	38	51.9	1.4	11	49	48.7	1.0
2	54	48.8	0.9	12	-	-	-
3	73	70.1	1.0	13	112	125.1	1.1
4	59	70.0	1.2	14	40	36.5	0.9
5	78	97.2	1.2	15	90	104.8	1.2
6	11	10.8	1.0	16	43	45.6	1.1
7	23	20.7	0.9	17	30	33.0	1.1
8	57	57.0	1.0	18	81	70.6	0.9
9	44	44.2	1.0	19	31	30.0	1.0
10	67	68.6	1.0	20	83	101.8	1.2
				Total	1063	1135.4	1.1

**Table 5 ijms-24-04077-t005:** Effects of the SNP markers associated with seed protein and oil content in the F_2:3_ population.

SNP	Chr.	LOD	Homo AA	Homo BB	R^2^ (%)
Protein					
Gm20_29512680	20	9.57	44.53	46.68	17.2
Oil					
Gm15_3621773	15	5.80	12.05	10.88	12.2

SNP, single nucleotide polymorphism; Chr, chromosome; LOD, likelihood of odds; Homo AA, Daepung allele; Homo BB, GWS-1887 allele.

**Table 6 ijms-24-04077-t006:** Effects of the SNP markers associated with seed protein and oil content in the BC_1_F_2:3_ population.

SNP	Chr.	LOD	Homo AA	Homo BB	R^2^ (%)
Protein					
Gm20_27578013	20	3.29	42.66	45.63	15.8
Oil					
Gm20_27578013	20	3.06	12.70	11.47	10.7

SNP, single nucleotide polymorphism; Chr, chromosome; LOD, likelihood of odds; Homo AA, Daepung allele; Homo BB, GWS-1887 allele.

**Table 7 ijms-24-04077-t007:** Markers and genotypes of BC_1_F_4_, two high-protein line selected from BC_1_F_3_ and advanced in generations.

Maker Name	BC_1_F_4_ Population No.																		
	141-1	141-3	141-5	141-6	141-7	141-8	141-10	141-12	141-14	145-1	145-3	145-4	145-5	145-7	145-9	145-10	145-11	145-14	145-15
BARC_1.01_Gm20_26500747	A ^†^	A	A	A	B	A	A	A	A	A	B	A	A	A	B	A	B	B	B
BARC_1.01_Gm20_26785339	A	A	A	A	B	A	A	A	A	A	B	A	A	A	B	A	B	B	B
BARC_1.01_Gm20_27578013	A	A	A	A	B	A	A	A	A	A	B	A	A	A	B	A	B	B	B
BARC_1.01_Gm20_28017701	A	A	A	A	B	A	A	A	A	A	B	A	A	A	B	A	B	B	B
BARC_1.01_Gm20_28070832	A	A	A	A	B	A	A	A	A	A	B	A	A	A	B	A	B	B	B
BARC_1.01_Gm20_28389137	A	A	A	A	B	A	A	A	A	A	B	A	A	A	B	A	B	B	B
BARC_1.01_Gm20_29512680	A	A	A	A	B	A	A	A	A	A	B	A	A	A	B	A	B	B	B
BARC_1.01_Gm20_29594697	A	A	A	A	B	A	A	A	A	A	B	A	A	A	B	A	B	B	B
BARC_1.01_Gm20_32603292	A	A	A	A	B	A	A	A	A	A	B	A	A	A	B	A	B	B	B
BARC_1.01_Gm20_32784352	A	A	A	A	B	A	A	A	A	A	A	A	A	A	A	A	A	A	A
BARC_1.01_Gm20_33001386	A	A	A	A	B	A	A	A	A	A	A	A	A	A	A	A	A	A	A
BARC_1.01_Gm20_33049242	A	A	A	A	B	A	A	A	A	A	A	A	A	A	A	A	A	A	A
BARC_1.01_Gm20_33580029	A	A	A	A	A	A	A	A	A	A	A	A	A	A	A	A	A	A	A
BARC_1.01_Gm20_33634187	A	A	A	A	A	A	A	A	A	A	A	A	A	A	A	A	A	A	A
BARC_1.01_Gm20_33727273	A	A	A	A	A	A	A	A	A	A	A	A	A	A	A	A	A	A	A
BARC_1.01_Gm20_33770596	A	A	A	A	A	A	A	A	A	A	A	A	A	A	A	A	A	A	A
BARC_1.01_Gm20_33842096	A	A	A	A	A	A	A	A	A	A	A	A	A	A	A	A	A	A	A
BARC_1.01_Gm20_34057184	A	A	A	A	A	A	A	A	A	A	A	A	A	A	A	A	A	A	A
Protein content (%)	44.40	44.81	45.47	43.90	47.58	44.32	43.38	43.60	42.94	45.87	50.17	45.55	45.66	43.92	47.43	45.24	48.60	48.04	48.48

^†^ ‘A’ genotype designates that the selected BC_1_F_4_–derived line was homogeneous for the allele from Daepung, ‘B’ designates that the line was homogeneous for the allele from GWS-1887.

**Table 8 ijms-24-04077-t008:** Statistics for the high-quality reads from GWS-1887 mapped to the reference soybean genome.

	GWS-1887
Total reads	260,761,086
Total size (bp)	39,374,923,986
Mapped reads (%)	98.8
Genome coverage (%)	95.3
Sequencing depth	38.6×
Number of SNPs	4,750,431
Number of InDels	897,005

**Table 9 ijms-24-04077-t009:** Candidate genes for seed protein content and InDels between the reference genome and GWS-1887.

Gene	Gene Position (bp)	Annotation	Position of Genetic Variation (bp)	GWS-1887
Glyma.20G082450	31,080,736..31,082,822	Ammonium transporter in embryo development	31082569	small indel
Glyma.20G084000	31,486,240..31,488,766	Small nuclear ribonucleoprotein F	31486478	missing allele
			31486484	missing allele
			31486498	missing allele
			31486737	ref
			31487037	ref
			31487188	ref
			31487573	ref
			31488329	ref
			31488590	ref
Glyma.20G084051	31,490,248..31,494,204	Protein FAR1-RELATED SEQUENCE 5-like	31490350	small indel
			31493215	small indel
Glyma.20G084100	31,538,100..31,541,615	Tetratricopeptide repeat (TPR)-like superfamily protein	31541401	small indel
			31541587	small indel
Glyma.20G084500	31,616,022..31,624,831	Pre-mRNA-processing factor 17-like isoform X1	31624215	small indel
Glyma.20G085100	31,724,592..31,729,626	CCT motif family protein	31727019	ref
Glyma.20G085250	31,749,953..31,750,702		31750696	small indel
Glyma.20G085300	31,772,592..31,776,697	Uncharacterized protein LOC100814338 isoform X3	31772869	small indel
			31775624	small indel
			31775992	large indel
Glyma.20G085400	31,778,024..31,782,508	60S ribosomal L23-like protein	31778202	small indel
			31778572	small indel
			31779221	small indel
			31779277	small indel
			31780211	small indel
			31780626	small indel
			31781580	small indel
			31782506	small indel
Glyma.20G085450	31,783,157..31,786,246	Uncharacterized protein LOC102669815	31783447	large indel
			31784986	small indel
Glyma.20G085500	31,799,904..31,802,040	HAD superfamily	31801435	small indel
Glyma.20G085700	31,931,514..31,933,982	Unknown protein	31931542	small indel
			31932782	small indel
			31933822	small indel
Glyma.20G085800	31,936,253..31,941,630	Eukaryotic translation initiation factor 4E	31940258	small indel
			31940854	large indel
Glyma.20G086100	31,963,336..31,965,943		31965594	ref
Glyma.20G086800	32,280,129..32,280,768	PA domain	32280386	small indel
Glyma.20g086900	32,344,475..32,346,588	Aldehyde dehydrogenase	32345329	small indel
			32345819	ref
			32346510	small indel
Glyma.20g087000	32,482,132..32,486,538	Signal transduction histidine kinase	32482188	small indel
			32482242	small indel
			32482250	small indel
			32482352	small indel
			32482410	small indel
			32482424	small indel
			32482464	small indel
			32482474	small indel
			32482480	small indel
			32482487	small indel
			32482549	small indel
			32482603	small indel
			32482697	small indel
			32482745	small indel
			32482761	small indel
			32482765	small indel
			32482811	small indel
			32482815	small indel
			32482850	small indel
			32482968	small indel
			32482972	small indel
			32482985	small indel
			32483011	small indel
			32483034	small indel
			32483036	small indel
			32483056	small indel
			32483091	small indel
			32483140	small indel
			32483171	small indel
			32483226	small indel
			32483233	small indel
			32483237	small indel
			32483278	small indel
			32483319	small indel
			32483325	small indel
			32483396	small indel
			32483411	large indel
			32483458	small indel
			32483552	small indel
			32485633	small indel
			32486435	small indel
Glyma.20G087600	32,693,094..32,698,696	Target SNARE coiled-coil domain protein	32694336	small indel
			32696653	small indel
Glyma.20G088000	32,881,065..32,886,678	S-adenosyl-l-methionine-dependent methyltransferases	32881614	small indel
			32882108	small indel
			32882111	large indel
			32882633	small indel
			32882909	small indel
			32883949	ref
			32883973	small indel
			32884174	small indel
			32884324	small indel
			32884601	small indel
			32885578	ref
			32885582	small indel
			32885642	ref
			32885647	ref
			32885839	ref
			32885841	small indel
			32885843	small indel
			32885941	small indel
			32886302	large indel
			32886312	small indel
			32886569	large indel
Glyma.20G088100	32,887,478..32,888,573	(S)-coclaurine N-methyltransferase-like	32887547	small indel
			32887824	ref
Glyma.20G088400	32,909,361..32,910,905	Oxidoreductase	32910800	small indel

## Data Availability

The original contribution presented in the study are publicly available.

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
