# Peer review of "Quantitative Trait Loci (QTL) Analysis of Seed Protein and Oil Content in Wild Soybean (Glycine soja)"

_ijms, 2023, doi:10.3390/ijms24044077_

Round 1

Reviewer 1 Report

In this study, the authors identified the QTL of soybean oil and protein. The research brings much information for researchers working in this field. However, the manuscri8pt should be revised before publication. The detailed information is listed as follows.

1. The authors used the population derived from a cross of a cultivated soybean Daepung 19 (Glycine max) with a wild soybean GWS-1887 (G. soja). However, the authors only emphasized the wild soybean soybean in the title. It might be nor reasonable.

2. The authors should not put the figures in the discussion section. They can put these figures in results section or supplementary datasets.

3. What is the difference of expression patterns of the two candidate genes Glyma.20g088000 (S-adenosyl-L-28 methionine-dependent methyltransferases) and Glyma.20g088400 (oxidoreductase, 2-oxoglutarate-29 Fe(II) oxygenase family protein) in two parental lines? Please add the expression pattern in seeds in two parental lines.

4. The format of the references should be improved. For example, line 372, “nature” should be “Nature”.

5. There are many errors in sentence punctuations, such as comma, semi- column, etc. The authors need to carefully check and make corrections throughout the manuscript to improve the English writing style for a scientific report.

Author Response

Response to Reviewer 1 Comments

Thank you so much for your suggestion. After considering this suggestion and reviewing the manuscript, we have amended the manuscript to your suggestion.

My comments and questions to authors are as follows:

Point 1: The authors used the population derived from a cross of a cultivated soybean Daepung 19 (Glycine max) with a wild soybean GWS-1887 (G. soja). However, the authors only emphasized the wild soybean soybean in the title. It might be nor reasonable.

Response 1: I understand the reviewer's point. The reason why we only emphasize wild soybean in the title is that the genes that specifically generate indels are derived from wild soybeans. This suggests that wild soybeans can be used to improve cultivated soybeans.

Point 2: The authors should not put the figures in the discussion section. They can put these figures in results section or supplementary datasets.

Response 2: Thanks to the reviewer's comments, the figure have been moved to the results section, and the manuscript was modified according to the comments.

Point 3: What is the difference of expression patterns of the two candidate genes Glyma.20g088000 (S-adenosyl-L-28 methionine-dependent methyltransferases) and Glyma.20g088400 (oxidoreductase, 2-oxoglutarate-29 Fe(II) oxygenase family protein) in two parental lines? Please add the expression pattern in seeds in two parental lines.

Response 3: I highly agree with the reviewer's key comments and thank you.

Regarding the expression patterns of the two genes, RNA expression analysis experiments are currently in progress, and follow-up papers will be published as soon as the experiments are completed.

Point 4: The format of the references should be improved. For example, line 372, “nature” should be “Nature”.

Response 4: The manuscript has been revised in consideration of the reviewer's comments.

Point 5: There are many errors in sentence punctuations, such as comma, semi- column, etc. The authors need to carefully check and make corrections throughout the manuscript to improve the English writing style for a scientific report.

Response 5: The manuscript has been revised in consideration of the reviewer's comments.

Reviewer 2 Report

The manuscript entitled: Quantitative trait loci (QTL) analysis of seed protein and oil content in wild soybean (Glycine soja), having authors: Woon Ji Kim, Byeong Hee Kang, Chang Yeok Moon, Sehee Kang, Seoyoung Shin, Sreeparna Chowdhury, Man-Soo Choi, Soo-Kwon Park, Jung-Kyung Moon and Bo-Keun Ha, presents that the levels of protein and oil are negatively correlated with each other and regulated by quantitative trait loci (QTL) that are controlled by several genes. Based on these results, two genes, Glyma.20g088000 (S-adenosyl-L-methionine-dependent methyltransferases) and Glyma.20g088400 (oxidoreductase, 2-oxoglutarate- Fe(II) oxygenase family protein), in which the amino acid sequence had changed and a stop codon was generated due to an InDel in the exon region, were identified. QTL was detected in the region of cqPRO-003, which has been previously reported as a major QTL related to protein content, but as a result of resequencing, no difference was observed from the recently cloned candidate gene cqPRO-003. On the other hand, new candidate genes Glyma.20g088000 and Glyma.20g088400, which contained InDels, were discovered. This suggests that the protein content may be regulated by the complex interaction of multiple genes and associations other than those that have previously been reported.

I consider that all the authors have an important contribution for this research and I consider that the manuscript can be published in International Journal of Molecular Sciences journal.

In the present study, the QTLs related to protein and oil content in the F2:3 population were identified as Gm20_29512680 and Gm15_3621773, respectively, whereas in the BC1F2:3 population, marker Gm20_27578013 was identified for both protein and oil. For more accurate confirmation of the location, crossovers around the QTL detected in BC1F2:3 were identified but could not be found, and it was concluded that a crossover occurred at markers Gm20_33049242 and Gm20_32603292 in the two BC1F3:4 lines, which was advanced one generation by selecting high-protein lines. Therefore, it was predicted that the protein-related gene is present in the region downstream of Gm20_32603292.It is interestingly to noticed that no mutation was detected in the Glyma.20G85100 gene of the CCT motif family protein, which was recently cloned as a major protein-related gene, and these results suggest that other major protein-related genes may be present in a similar genetic region.

The crude protein content was measured using the Kjeldahl method using a Kjeltec 1030 Autoanalyzer, while the crude oil content was measured using ether extraction, both being methods well known for this type of analysis. The design of the experiments is well-done, in order to obtain a better explaination of the results.

The data presented in this paper are very carrefull selected and presented. The data are scientific explained.

The paper is well written and presented, for all the scientific data.

I recommend the paper Quantitative trait loci (QTL) analysis of seed protein and oil content in wild soybean (Glycine soja), for being published in International Journal of Molecular Sciences journal.

Author Response

Thanks for the reviewer's comments.

Parts of the manuscript have been added or modified.

Round 2

Reviewer 1 Report

The authors have replied all my concerns.